# OpenReview forum: "Multilingual and  Multi-Accent Jailbreaking of Audio LLMs"
_colmweb.org/COLM/2025/Conference — COLM 2025_

### Official Review · Reviewer_57ot · 2025-05-12

**Rating:** 7
**Confidence:** 4
**Ethics Flag:** 1

**Summary:**

The authors evaluate the robustness of audio language models (ALMs) to multlingual and multi-accent jailbreaks. The authors explore a little bit about defending against said jailbreaks.

**Reasons To Accept:**

- The experiments are cleanly described
- The experiments are thorough

**Edit 2025/06/09: The authors' responses convinced me that their paper is making more of a conceptual contribution than I originally thought. They addressed my reasons to reject and while I think the paper is incremental, the work seems solid and correct, so I'm upgrading my score to a 7. I'm concerned that the score of 8 by Reviewer nssE is undeserved and I thought about lowering my score to 6 to compensate but decided to instead leave it to the AC to sort this out.**

**Reasons To Reject:**

- My main objection to this paper is that I feel like it doesn't add much conceptually to our knowledge of multimodal language models. We know that LMs, VLMs, ALMs can be jailbroken. We know that perturbations to inputs can increase attack success rates (ASRs). We specifically know that ALMs can be jailbroken via audio inputs, and can be jailbroken via multi-accent perturbations. We know that additional modalities increase the attack surface of multimodal models. I'm left wondering what this paper contributes other than more systematically evaluating multilingual audio attacks.

- Figure 2: There should be confidence intervals based on the number of samples in the benchmark. My understanding is that there are 520 harmful instructions. If so, can you please indicate the uncertainty with the scores so the reader can evaluate which are meaningfully different?

- Line 248: This entire paragraph is speculation. Perhaps move it to Discussion, or add experiments testing this? This content seems misplaced in Section 5 Results.

- Section 5.4.: Related to the above point, if the authors are interested in defending against such audio jailbreaks, why not evaluate safety finetuning? For example, take AdvBench and use the audio messages to train the model to refuse such harmful requests. This seems relatively straightforward to implement and evaluate.

- The insight that "attackers can exploit the weakest link in a multimodal system [...] ,highlighting how increased model flexibility (e.g., more parameters, diverse input modalities) inadvertently introduces greater attack surfaces" is well known, especially in the vision-language model research lineage.

- Figure 1: Please fill out the caption. There's a lot happening in the image and the caption wholly fails to communicate the contents.

- nit: I think `Multi-AudioJail` is a bit of an awkward name. Perhaps consider extending it to be more descriptive, such as including the full "Jailbreak", or if that's too long, using an acronym

---

> ### Author Response · Authors · 2025-06-02
>
> **Response to Reasons to Reject 6:**  Thank you for the suggestion. We will expand the caption in the following manner:
>
> Overview of MULTI-AUDIOJAIL attack framework. a) **Text-only Attack (Multilingual)**: English jailbreak prompts are translated into multiple target languages, converted to speech via TTS, and passed to an audio LLM — yielding low jailbreak success.  b) **Audio-Only Attack (Multilingual / Multi-Accent): Raw audio prompts in different languages or accents (both native and synthetic) are fed directly into the LALM; some language/accent bypass safety filters while others still trigger refusal.  c) Audio Perturbation Attack (Multilingual / Accent): Multilingual and multi-accent audio inputs are further modified with adversarial acoustic perturbations before inference, dramatically increasing jailbreak success across all variations.
>
> ---
>
> **Response to Reasons to Reject 7:**  Thank you for the suggestion. We will consider renaming our framework to **Multi-AudioJail (MAJ)** for clarity, updating all references accordingly.

---

> ### Author Response · Authors · 2025-06-02
>
> We appreciate Reviewer 57ot’s thorough reading and constructive feedback on our submission. Below, we address each of your concerns in turn.
>
> **Response to Reasons to Reject 1:** Thank you for your insights. We agree that jailbreak vulnerabilities in other modalities are well-documented. However, our work goes beyond re-evaluation by introducing several novel and practical contributions that deepen our understanding of multimodal vulnerabilities—particularly in real-world, non-adversarial settings. First, we show that benign, naturally spoken prompts—across languages, accents, and environments—can significantly raise JSRs. These are realistic utterances likely to occur in deployed audio systems, making our findings directly actionable.  Second, we identify a new audio-specific attack vector: combining multilingual/accent speech with natural acoustic perturbations. These arise organically (e.g., in public spaces or phone calls) and require no technical expertise, unlike traditional attacks. Third, we present the first benchmark dataset for adversarial multilingual/accent audio jailbreaks, evaluating five state-of-the-art LALMs, 7 accents, and 13 multiple languages—all evaluated using fully open-source models to ensure reproducibility and practical applicability. The perturbed inputs remain semantically coherent, as evidenced by high SQA accuracy and consistently unsafe outputs flagged by LLaMa-Guard.
>
> ---
>
> **Response to Reasons to Reject 2:**  Thank you for the suggestion. We have added 95% confidence intervals (CIs) using normal approximation to the binomial proportion (Table 1 below). These will be included in the revised Figure 2 with three additional languages. Notably, CIs for audio and text often do not overlap—indicating statistical differences.
>
> Table 1:
> | Language   | Audio JSR (%) | 95% CI (Audio)     | Text JSR (%) | 95% CI (Text)     |
> |------------|---------------|--------------------|---------------|-------------------|
> | English    | 2.02          | [0.81%, 3.23%]     | 2.38          | [1.07%, 3.69%]    |
> | French     | 4.79          | [2.95%, 6.63%]     | 3.53          | [1.94%, 5.12%]    |
> | Spanish    | 5.44          | [3.49%, 7.39%]     | 3.15          | [1.65%, 4.65%]    |
> | German     | 12.31         | [9.49%, 15.13%]    | 3.92          | [2.25%, 5.59%]    |
> | Italian    | 6.54          | [4.42%, 8.66%]     | 4.07          | [2.37%, 5.77%]    |
> | Portuguese | 6.29          | [4.20%, 8.38%]     | 6.13          | [4.07%, 8.19%]    |
>
> ---
>
> **Response to Reasons to Reject 3:**  We will move that paragraph to a new section of Discussion or combine with Section 7 as Discussion and Limitation. We will properly label it as a hypothesis in Section 5, reserving it for the discussion of future work.
>
> ---
>
> **Response to Reasons to Reject 4:** Thank you for the suggestion. We agree that safety fine-tuning could strengthen defenses, but we focused on a zero-shot inference-time defense for its practicality. It avoids retraining, integrates via prompt modification, and aligns with workflows supported by APIs like OpenAI's [1].  Our method requires no new data and avoids risks tied to safe data quality. As noted in our response to Reviewer uBUp, our defense preserves utility: <2% drop in English SQA accuracy across 5 models. This plug-and-play method provides a lightweight safeguard for LALMs. That said, we view safety fine-tuning as a promising complementary direction for future work.
>
> **Reference:**
> [1] OpenAI. *“Audio API Guide.”* https://platform.openai.com/docs/guides/audio
>
> ---
>
> **Response to Reasons to Reject 5:**  Thank you for the thoughtful comment. We agree that the general idea has been noted in prior work. However, our contribution is the first to systematically demonstrate this effect in the **audio modality**, which is increasingly central to real-world AI systems (e.g., voice assistants, multimodal agents). To clarify this distinction, we will revise the sentence in Section 1. Specifically, we show that even a human-intelligible audio perturbations can meaningfully compromise model safety, establishing audio as a standalone channel of vulnerability. As multimodal models expand in deployment, our findings underscore the importance of evaluating each input modality not in isolation, but as a possible attack vector. We view this as a timely and necessary step toward safeguarding next-generation multimodal systems, with audio representing a critical yet underexplored dimension.

---

### Official Review · Reviewer_nGKb · 2025-05-12

**Rating:** 6
**Confidence:** 4
**Ethics Flag:** 1

**Summary:**

This paper introduces Multi-AudioJail, a framework for exploring effects of adversarial multilingual, multi-accent, and acoustic perturbations on jailbreaking large audio language models (LALMs). The authors created a new dataset by selecting 520 harmful instructions from AdvBench (in English), translating them into 5 other languages (German, Italian, Spanish, French, and Portuguese), then using TTSMaker to create audio prompts. Two multi-accent configurations were studied for English prompts: natural accents (audio generated from TTS models trained on accented English) and synthetic accents (English texts fed to models originally trained for other languages including Chinese, Korean, Japanese, Arabic, Portuguese, Spanish, and Tamil). Besides clean audio prompts, the 5 adversarial acoustic conditions were applied: 3 types of reverberations, echo, and and whisper effects.

Using this dataset, the authors then evaluate 5 recent LALMs: MERaLiON, MiniCPM, Qwen2, DiVa, and Ultravox. The main metric of interest is jailbreak success rate (JSR), where a response is classified as safe/unsafe and the LAML refuses to answer. This was done using LLaMa Guard 3 and a small subset was validated by human annotators. The results show that the proposed perturbations significantly increased JSRs across the board.

**Questions To Authors:**

Minor comments/typos:
- Line 209: typo "China" --> "Chinese"

**Reasons To Accept:**

1. The paper introduces a new framework and dataset for assessing LAML vulnerabilities, which is valuable to the growing audio- and multimodal-LM research area.
2. The data creation pipeline is clear and reproducible.
3. The experiments are quite thorough and done across multiple recent LAMLs.
4. The paper is clear and easy to follow; key findings are clearly presented and emphasized.

**Reasons To Reject:**

My main concern is regarding the effects of poorly generated audio vs. true jailbreaking success, i.e. how do you tease apart whether a model refuses a prompt because the audio is so bad it is no longer intelligible vs. model refusing a prompt because it is truly unsafe? For this, there are some experiments I can think of (and maybe the authors already have the answers to):
- Using safe audio inputs but with the same accent/acoustic perturbations and see if the LAML refusal rates are significantly different
- Compare JSRs across WER bins, i.e. it's likely that the higher the WER, the higher JSRs. Ideally this would be done for WER of the individual LAMLs, *not* the common Whisper-large-v3 system, since individual WERs might work as a loose proxy for each model's ability to "correctly" process an input audio's content
- The claim regarding how "acoustic perturbations interact with cross-lingual phonetics to cause JSRs to surge..." (in Abstract) is not fully expounded on, since the experiment results all focus on JSRs and there are no explanations or analyses on the type of "cross-lingual phonetics interaction" claimed

Other minor concerns/questions I have are regarding the variety of languages and perturbations:
- The languages translated to are all western-centric, I assume this is limited by available and reliable translation services
- For the synthetic accents setting, did you listen to the TTS generated samples? Did they sound intelligible? What are the (clean) WERs for the synthetic accents? I think Table 6 only provides WERs for perturbed samples?
- The 3 main acoustic perturbation types seem a bit arbitrary even though they are reasonable - what was the motivation for choosing these? In speech enhancement/speech quality assessment research, usually the first and most common distortions studied are some types of additive background noise - I wonder if you have experimented with these? (See for example https://urgent-challenge.github.io/urgent2025/)

---

> ### Author Response · Authors · 2025-06-01
>
> **Response to Reasons to Reject 4:** We focused on Western languages because current LALMs tend to handle them more reliably. In our preliminary trials with non-Western languages (e.g., Chinese, Korean, Arabic), models often produced incorrect or off-topic outputs—likely due to limited multilingual training. Additionally, high-quality TTS was inconsistent for several of those languages. To ensure fair and consistent evaluation, we selected languages with robust model support and TTS quality. Expanding to more diverse languages is an important direction we plan to explore as model and TTS capabilities improve.
>
> ---
>
> **Response to Reasons to Reject 5:** Thank you for the thoughtful question. Yes, we manually reviewed all TTS-generated synthetic accent audio samples to confirm their intelligibility. Based on this inspection, we verified that all clean synthetic samples were intelligible to human listeners. To further support this, we will include example audio clips for each perturbation type in our supplementary materials.
>
> We computed average WERs for clean synthetic and natural accents using Whisper-large-v3. As shown in the table below, these inputs are highly intelligible in unperturbed form. While not model-specific, these WERs serve as a global proxy for intelligibility, since LALMs operate in end-to-end QA mode without exposing transcriptions (see Response to Reason to Reject 3). Importantly, JSRs remain low on clean inputs but increase sharply under perturbations, indicating the models understand the harmful content but fail to apply safety constraints. If the audio had been unintelligible, outputs would have been off-topic and flagged as “safe” by LLaMa-Guard. These findings reinforce our conclusion: LALMs preserve semantic understanding under perturbation, and high JSRs reflect safety failures—not transcription errors.
>
> | Type       | Accent             | Avg. Clean WER |
> |------------|--------------------|----------------|
> | **Synthetic**  | Arabic             | 0.6903         |
> |            | Spanish (Spain)    | 0.3687         |
> |            | Japanese           | 0.7686         |
> |            | Korean             | 0.4472         |
> |            | Portuguese         | 0.3336         |
> |            | Tamil (India)      | 0.1645         |
> |            | Chinese (Mandarin) | 0.0971         |
> |------------|--------------------|----------------|
> | **Natural**    | Australia          | 0.0983         |
> |            | India              | 0.0933         |
> |            | Kenya              | 0.0956         |
> |            | Nigeria            | 0.0939         |
> |            | Philippines        | 0.0928         |
> |            | Singapore          | 0.1009         |
> |            | South Africa       | 0.0961         |
> |            | United Kingdom     | 0.0972         |
>
> ---
>
> **Response to Reasons to Reject 6:**  There could be many such perturbations since audio modality allows unlimited flexibility including what we did and background noise/chatter but we wanted to focus on the few perturbations and explore them in-depth. Therefore, our work establishes a lower bound on safety vulnerability of LALMs. We would also like to emphasize that one of our motivations was to explore how even seemingly benign environmental sounds can act as unintentional adversarial perturbations, without malicious intent from the user. To further expand on this, we will consider incorporating our own approach of blending background noises with diverse accents and languages, which could reveal novel interactions between naturalistic acoustic interference and linguistic variation. This would allow us to build on past work while offering a unique angle aligned with our multilingual attack focus.
>
> ---
>
> **Response to Reasons to Reject 7:** Thank you for the correction. We will proofread and address any additional copy edits.

---

> ### Author Response · Authors · 2025-06-01
>
> We thank the reviewer for the thoughtful critique and for recognizing the value of our Multi-AudioJail framework and dataset. Below, we address each of your concerns and outline planned revisions
>
> **Response to Reasons to Reject 1:** Thank you for highlighting the importance of distinguishing between refusals due to unintelligible audio and those caused by true safe detection. To address this, we conducted an evaluation using perturbed safe audio inputs with the same accent and adversarial settings. Our goal is to verify whether the model could still understand and respond appropriately to benign queries when the same distortions were applied.
>
> As shown in the table below, SQA accuracy on both Whisper and Reverb Teisco inputs remained high across all models and the three example accents shown. This suggests that models retain core utility under perturbation (full results will be included in the final version of our paper). These results reinforce that LALMs still understand benign content under realistic distortions, and elevated JSRs in harmful queries cannot be attributed to intelligibility loss. Our results confirm that jailbreaks stem from genuine safety vulnerabilities in LALMs under realistic input conditions and not from degraded audio.
>
> | **Model** | **Accent**     | **Clean (%)** | **Whisper (%)** | **Reverb Teisco (%)** |
> | --------- | -------------- | ------------- | --------------- | ---------------------- |
> | MERaLiON  | Australia      | 95.0          | 97.0            | 87.0                   |
> |           | India          | 98.0          | 94.0            | 86.0                   |
> |           | Nigeria        | 94.0          | 91.0            | 85.0                   |
> | MiniCPM   | Australia      | 94.0          | 96.0            | 83.0                   |
> |           | India          | 96.0          | 94.0            | 81.0                   |
> |           | Nigeria        | 94.0          | 95.0            | 79.0                   |
> | Qwen2     | Australia      | 79.0          | 77.0            | 64.0                   |
> |           | India          | 85.0          | 77.0            | 58.0                   |
> |           | Nigeria        | 84.0          | 76.0            | 62.0                   |
> | Ultravox  | Australia      | 72.0          | 63.0            | 82.0                   |
> |           | India          | 80.0          | 74.0            | 73.0                   |
> |           | Nigeria        | 79.0          | 68.0            | 75.0                   |
> | DiVA      | Australia      | 92.0          | 90.0            | 73.0                   |
> |           | India          | 89.0          | 85.0            | 72.0                   |
> |           | Nigeria        | 91.0          | 89.0            | 75.0                   |
>
> ---
>
> **Response to Reasons to Reject 2:** Thank you for the thoughtful suggestion. We agree that analyzing the relationship between WER and JSR is valuable for assessing whether high JSRs are due to transcription errors. Our results show that acoustic perturbations preserve utility while significantly increasing vulnerability:
> * As shown in Table 5, Echo and Whisper produce low WERs (0.108 and 0.102) but significantly increase JSRs (e.g., +42.21 for German under Whisper, Table 13), indicating the model comprehends the input rather than failing to transcribe.
> * We also followed  [1, 2] to force transcription via explicit instructions (e.g., "Transcribe the input audio."), but these consistently failed when the input was phrased as a question. In such cases, models either answered semantically or refused due to safety filters.
>
> This highlights a key challenge in speech QA: transcription and response generation are tightly coupled. If models failed to understand the input, outputs would be off-topic and marked “safe.” Instead, coherent but unsafe responses confirm that models comprehend the content but fail to enforce safety.
>
> **References:**
>
> [1] https://huggingface.co/MERaLiON/MERaLiON-AudioLLM-Whisper-SEA-LION
>
> [2] https://huggingface.co/openbmb/MiniCPM-o-2_6
>
> ---
>
> **Response to Reasons to Reject 3:** We understand that our phrasing on “cross-lingual phonetics interaction" may have introduced ambiguity.  Our intention was to emphasize that applying acoustic perturbations to multilingual/accent audio inputs substantially increase JSRs. To clarify this point, we will revise the sentence in the abstract as follows:
> * *“…a hierarchical evaluation pipeline revealing that applying acoustic perturbations (e.g., reverberation, echo, whisper effects) to multilingual and multi-accent audio inputs significantly amplifies jailbreak success rates (JSRs) ...*

---

> > ### Comment · Reviewer_nGKb · 2025-06-08
> >
> > Thank you for the answers and clarifications.
> >
> > **Regarding Response 1**: I'm not sure I agree that a 10% drop in SQA accuracy for Reverb Tesco inputs (pretty much across the board) is negligible to claim that therefore "elevated JSRs in harmful queries cannot be attributed to intelligibility loss." I interpret this as perturbation does significantly affect intelligibility of benign audios (unsurprisingly).
> >
> > **Regarding the remaining comments**: the responses make sense to me.

---

> > > ### Author Response · Authors · 2025-06-09
> > > **Response to Reviewer nGKb’s Comment on Response 1**
> > >
> > > Thank you for this thoughtful follow-up and for pointing out the SQA accuracy drop under Reverb Teisco. We agree that a ~10% point decline in benign question answering reflects a non-negligible impact on intelligibility, and we will revise our wording accordingly to better acknowledge this limitation.
> > >
> > > That said, we believe our central claim—that LALMs exhibit genuine safety vulnerabilities under realistic perturbations—still holds for the following reasons:
> > >
> > > 1. Even under the strongest perturbation (Reverb Teisco), average SQA accuracy remains above 70% across most models and accents, with several cases exceeding 80%. This suggests that the models still successfully comprehend benign spoken content to a certain extent despite acoustic degradation. While our SQA set is distinct from VoiceBench [1], it's worth noting that top performing models in VoiceBench (e.g., GPT-4o-Audio, Whisper+GPT-4o) achieve similar SD-QA scores in the 70-75% range on a different but similarly realistic benchmark. Additionally, since our evaluation uses a relatively small set of 100 questions, the observed ~10% drop may be amplified by sampling variance.
> > >
> > > 2. While benign comprehension drops by ~10%, the corresponding increase in JSRs for harmful prompts is significantly larger—often in the range of 30-50% points. This discrepancy suggests that the rise in unsafe completions is not simply a byproduct of degraded input quality but rather a result of robustness failures in safety enforcement.
> > >
> > > 3. In most cases, the responses to perturbed harmful prompts remain coherent and contextually aligned, which would be unlikely if the models had failed to understand the input. If so, Llama Guard would have automatically neglected and classified as a "safe" content, not contributing to the JSR. This further supports the claim that LALMs can comprehend the queries but still fail to apply appropriate safety filters under acoustic perturbation.
> > >
> > > Overall, we sincerely appreciate the reviewer's constructive comment and will incorporate a more nuanced discussion of intelligibility vs. safety failure into Section 4.1 of our paper.
> > >
> > > [1] VoiceBench leaderboard: https://github.com/MatthewCYM/VoiceBench

---

### Official Review · Reviewer_nssE · 2025-05-12

**Rating:** 7
**Confidence:** 3
**Ethics Flag:** 2

**Summary:**

This paper is among the first to explore multilingual and multi-accent jailbreaking of Audio LLMs. They present a novel dataset of adversarially perturbed multilingual and multi-accent audio jailbreaking prompts and also offer an evalution pipeline to reveal the vulenerabilites of current multimodels and unimodel. They also found that multimodels are eaier to jailbreak than unimodel with 3.1x higher attack success rate. They also promise to release the datasets for the community to use.

**Ethics Concerns Details:**

the audio jailbreaking might be offensive to some audience.

**Questions To Authors:**

considering the uniqueness of audio, what's the fundamental difference between jailbreaking techniques for audio, image or text?

**Reasons To Accept:**

1. The importance of audio jailbreaking with  multilingual and multi-accent inputs are overlooked and this paper presents a good starting point to explore this direction. I would expect more research in this direction.

2. The paper conducted a series of comprehensive experiments for this task and also present a lot of analysis and comparison on different models, languages, methods, which is valuable for the community.

**Reasons To Reject:**

1. It would be great if the authors can provide some examples as supplemental materials to show how the audio sounds as case studies.

2. I am not very familiar with Audio domain so my reivew could be biased. I lowered my confidence score.

---

> ### Author Response · Authors · 2025-05-31
>
> We sincerely appreciate **Reviewer nssE’s** comments on our paper and we would like to address the reviewer’s concerning points as following:
>
> **Reasons to Reject 1:** *“It would be great if the authors can provide some examples as supplemental materials to show how the audio sounds as case studies.”*
>
> **Response to Reasons to Reject 1:** Thank you for the suggestion. We will upload supplementary audio examples of each perturbation scenario in the final version of our paper, allowing the reviewers to hear the perturbed samples directly.
>
> ---
>
> **Question 1:** *“Considering the uniqueness of audio, what's the fundamental difference between jailbreaking techniques for audio, image or text?”*
>
> **Response to Question 1:**  Unlike text or images — where adversarial jailbreaks rely on carefully crafted prompts or pixel-level perturbations generated offline — audio introduces an analog channel of attack parameters that can be *manipulated in real time*. In our work, we demonstrate that simply altering a speaker’s accent or introducing natural acoustic effects (e.g., reverberation, echo, background noise, whispering) is sufficient to push a “safe” audio LLM out of its alignment distribution and trigger unsafe behaviors.
>
> These perturbations are fundamentally different because:
>
> - *They occur naturally* (e.g., regional accents, tunnel echoes) and do *not require model access or optimization*;
> - *They can be applied dynamically during live conversations* with voice assistants, making them harder to detect or pre-filter;
> - *They exploit the continuous, time-varying nature of sound*—creating a far richer attack surface than the *discrete tokens* used in text or *static pixels* in images.
>
> By leveraging **analog acoustic manipulations** that any user or man-in-the-middle device could introduce, we reveal a novel and practical class of jailbreaks that is unique to the audio modality.

---

> > ### Comment · Reviewer_57ot · 2025-06-01
> > **Follow up Question to Authors' Response to Reviewer nssE**
> >
> > I happened to read the below sentence and I was hoping you could provide some additional clarification for me.
> >
> > >  Unlike text or images — where adversarial jailbreaks rely on carefully crafted prompts or pixel-level perturbations generated offline — audio introduces an analog channel of attack parameters that can be manipulated in real time.
> >
> > What does "manipulated in real time" mean in this context? My understanding from Section 3.2.2. was that the perturbations were applied to the original clean recordings offline. Or were you trying to communicate that audio _could_ be manipulated in real time even though it _isn't_ manipulated in real time in this paper?

---

> > > ### Author Response · Authors · 2025-06-01
> > >
> > > Thank you for your thoughtful question, and we appreciate the opportunity to clarify.
> > >
> > > You are correct in observing that in our experiments (Section 3.2.2), the perturbations were applied offline to pre-recorded clean audio inputs. Our current evaluation is conducted in a controlled, reproducible setting where all acoustic perturbations are synthetically applied and saved before inference.
> > >
> > > When we state that audio introduces an *"analog channel of attack parameters that can be manipulated in real time"*, we are referring to the theoretical and practical feasibility of applying such perturbations dynamically during live use, even though our study does not do so.
> > >
> > > For example:
> > > * **Whispering**, speaking in an accent, or moving into a **reverberant space** (like a tunnel or bathroom) are actions that a user could naturally perform during a live interaction with LALMs.
> > > * **A man-in-the-middle device** (e.g., a smart speaker or microphone relay) could introduce signal-level effects like echo or room impulse response convolution in real time without needing access to the model or retraining.
> > >
> > > Thus, while our implementation is offline, the attack vector we evaluate inherently real-time capable and non-reliant on optimization or prior model knowledge, which contrasts with the prompt engineering or gradient-based attacks often used in text/image domain.
> > >
> > > We will rephrase the sentence in the final version of the paper to make this distinction clearer. Thank you again for catching this and helping improve the precision of our claims.

---

> > ### Comment · Reviewer_nssE · 2025-06-11
> >
> > Thanks. I am not very familiar with Audio domain so my reivew could be biased. I lowered my confidence score.

---

> ### Comment · Reviewer_57ot · 2025-06-01
> **Reviewer nssE - Aligning On Our Scores**
>
> Hi Reviewer nssE,
>
> I've read through your review for Submission 612 and wanted to touch base on a few points to ensure we have a shared understanding.
>
> - Regarding the paper's novelty, your review suggests the area of multilingual/multi-accent audio jailbreaking is "overlooked" and positions this as a "good starting point." The authors, however, cite multiple prior works in this domain on Page 1 and in Section 2.2. I'm trying to reconcile these observations – could you clarify how you see this paper's contribution as a foundational "starting point" in that context?
>
> - You positively noted the "comprehensive experiments." To help me see it from your perspective, could you highlight which aspects of the experimental setup, analysis, or specific results you felt were particularly strong or insightful? The review doesn't go into these details, and I think it would be valuable for the discussion.
>
> - The "Reasons To Reject" focuses on adding audio examples. While a good suggestion, this seems more like a minor revision. Do you have any actual reasons to reject?
>
> Your overall assessment is extremely positive (8/10, 5/5 confidence), and I'm not seeing the strengths you've identified. A bit more detail on your reasoning for these points would be very helpful.
>
> Thank you!

---

> > ### Comment · Reviewer_nssE · 2025-06-11
> >
> > Thanks for following up. I am not very familiar with Audio domain so my reivew could be biased. I lowered my confidence score.

---

### Official Review · Reviewer_uBUp · 2025-05-27

**Rating:** 6
**Confidence:** 3
**Ethics Flag:** 1

**Summary:**

The paper proposes jailbreak attacks on audio LLMs leveraging multi-lingual and multi-accented inputs combined with audio perturbations such as echo, reverberation, and whisper increases attack success rates by more than 50% in multiple languages and accents. The experiments analyze 5 audio LLMs with low attack success rates in text-only inputs, and modify the inputs using 6 languages, 6 natural accents, and 7 synthetic accents using TTS API from Azure. Defense techniques like inference-time prompting with exemplars help reduce the attack success rate.

**Reasons To Accept:**

* The attacks are extensive in nature with multiple languages and accent-types used in generating attacks. This along with the audio perturbations improve the attack success rates in most audio LLMs
* The ablation rates varying the delay and decay factor illustrate that the attack success can be controlled based on the audio modulations added to the jailbreaks.

**Reasons To Reject:**

* The reduction in word error rates (WER), and SQA accuracy in multilingual and multi-accent attacks indicate that the utility of the audio LLMs is also reduced with the increase in attack success rates. This could indicate that these input distributions are out-of-distribution for all types of inputs - both benign and adversarial.
* Hence, the increased attack success rates could indeed be in distributions where the model has low utility and hence may not be usable in downstream applications. This analysis of high attack success rates in data distributions with high utility needs to be explored to further make this finding compelling.
* In evaluating the defense, applying the same attacks on benign prompts would also reduce the utility of the prompts - this needs to be measured and contrasted with the jailbreak attacks to understand the efficacy of the defense proposed as the instructions would uniformly apply to benign prompts as well.

---

> ### Author Response · Authors · 2025-06-01
>
> **Reasons to Reject 1:** *“The reduction in word error rates (WER), and SQA accuracy in multilingual and multi-accent attacks indicate that the utility of the audio LLMs is also reduced with the increase in attack success rates. This could indicate that these input distributions are out-of-distribution for all types of inputs - both benign and adversarial.”*
>
> **Response to Reasons to Reject 1:**
> Thanks for the suggestion and we agree that adversarial prompts are often out-of-distribution (OOD) relative to the model’s safety alignment data. Indeed, prior works [1, 2] have shown that simple transformations (e.g., mixup, visual/textual blending) applied to harmful prompts can “OOD-ify” them, thereby bypassing safety filters. These findings reinforce a key observation in jailbreak literature: many failures stem from the distributional gap between safety-aligned inputs and adversarial ones.
>
> Our work builds on this by examining various adversarial audio inputs. Importantly, our experiments show that LALMs retain meaningful comprehension of harmful queries even after perturbation. This is because the resulting responses are coherent and contextually relevant — and consistently flagged as “unsafe” by LLaMa-Guard. If the inputs had been incomprehensible, the model’s outputs would have been off-topic and thus automatically filtered as “safe”. Therefore, the elevated JSRs point to failures in safety enforcement, not degradation of language understanding.
>
> **References:**
> [1] Jeong et al. *“Playing the Fool: Jailbreaking LLMs and MLLMs with out-of-distribution strategy”*
> [2] Peng et al. *“Logic Jailbreak: Efficiently Unlocking LLM Safety Restrictions through Formal Logical Expressions”*
>
> ---
>
> **Reasons to Reject 2:** *“Hence, the increased attack success rates could indeed be in distributions where the model has low utility and hence may not be usable in downstream applications. This analysis of high attack success rates in data distributions with high utility needs to be explored to further make this finding compelling.”*
>
> **Response to Reasons to Reject 2:**
> Thank you for raising this important point. While adversarial queries are indeed out-of-distribution from the perspective of the model’s safety alignment, our findings show that audio manipulations preserve the general utility of LALMs even under attack settings.  Specifically, transformations such as echo, whisper, and Teisco reverberation result in a low WER of less than 0.26 across six languages (Table 5), while SQA accuracy for benign prompts remains between 8% and 98% (avg. 67.4%) across five non-English languages (Table 7). Despite this, the same perturbations increase JSRs by 27–48 percentage points (Table 1).
>
> This confirms that the models continue to process and understand the input audio effectively, even in perturbed conditions. The rise in attack success is not due to low utility or input degradation, but rather due to breakdown in safety filtering while preserving the harmful intent in acoustically altered inputs.
>
> ---
>
> **Reasons to Reject 3:** *“In evaluating the defense, applying the same attacks on benign prompts would also reduce the utility of the prompts - this needs to be measured and contrasted with the jailbreak attacks to understand the efficacy of the defense proposed as the instructions would uniformly apply to benign prompts as well.”*
>
> **Response to Reasons to Reject 3:**
> Thank you for your suggestion. To quantify the impact of our defense on benign audio queries, we measured English SQA accuracy with our defense. The models achieved 88.0% (DiVA), 97.0% (MeraLiON), 96.0% (MiniCPM), 85.0% (Qwen2), and 84.0% (Ultravox), which is < 2% change compared to corresponding clean-inputs baseline. This demonstrates that our defense does not materially degrade benign SQA performance.
>
> ---

---

### Decision · Program_Chairs · 2025-07-08

**Decision:**

Accept

**Comment:**

The paper represents a reasonable step forward in the space of audio jailbreaking, with new data and improved success rates. While there was some concern about the quality of the audio after translation and conversion and whether the ASR is due to quality issues or not, the authors presented follow-up evidence that quantified this delta. While this confirms that the jailbreak evaluation is due in part to a drop in audio quality, some amount of performance drop is reasonably to be expected and is not grounds for rejection as long as it is documented properly. The authors should make this ablation and breakdown of the improved ASR with respect to quality vs accent/lingual properties clear in the final version.

**This paper went through ethics reviewing. Please review the ethics decision and details below.**
Decision: All good, nothing to do  or only minor recommendations